# Beyond the jab: Unravelling the complexities of vaccine adoption for East Coast Fever in rural Kenya

Ann W. Muthiru[1]*, Josphat Muema[2,3,4], Nyamai Mutono[3,4], S. M. Thumbi[3,5], Salome A. Bukachi[1,6]

1 Department of Anthropology, Gender and African Studies, University of Nairobi, Nairobi, Kenya, 2 Feed the Future Innovation Lab for Animal Health, Washington State University, Pullman, WA, United States of America, 3 Centre for Epidemiological Modelling and Analysis, University of Nairobi, Nairobi, Kenya, 4 Washington State University Global Health Program–Kenya, Nairobi, Kenya, 5 Paul G. Allen School for Global Health, Washington State University, Pullman, WA, United States of America, 6 Department of Anthropology, Durham University, Durham, United Kingdom

* mmuthiruann@gmail.com

**Funding:** This work was funded in whole or in part by the United States Agency for International Development (USAID) Bureau for Resilience and Food Security under Agreement #

## Abstract

East Coast Fever (ECF) is one of the leading causes of livestock mortality and reduced productivity across Eastern Africa, and while a live vaccine against it known as the Infection and Treatment Method has existed for three decades now, its adoption by affected communities remains low. This study sought to provide a detailed examination of the dynamics that shape Infection Treatment Method (ITM) vaccine adoption behaviours. The study examined individual, socio-cultural and ecological- level factors influencing ITM adoption using the socio-ecological model. Analyzing data obtained from 18 focus group discussions, 30 in-depth interviews with livestock keepers, and 25 key informant interviews conducted with community stakeholders, the study identified factors associated with vaccine adoption within pastoralist communities in rural Kenya. These factors included knowledge and awareness of the Infection Treatment Method vaccine, its cost, livestock keepers' perceptions of East Coast fever relative to other livestock diseases, wildlife-livestock interactions, climate as contributing factors, and wildlife-livestock interactions influencing ECF risk and severity. Overall, the study findings emphasize the need for multifaceted strategies to increase vaccine adoption among livestock keepers.

## Introduction

East Coast Fever (ECF) is an infectious tick-borne disease of cattle caused by *Theileria parva (T.parva)*, a parasite transmitted through brown ear ticks *(Rhipicephalus appendiculatus)*. This devastating disease, particularly prevalent throughout Africa's central, eastern, and southern parts, is estimated to account for over $300 million annually in losses, attributed to mortality, decrease in milk production, lower fertility, poor condition of the cattle associated with lower value at the market and treatment costs [1, 2]. For many livestock-dependent communities, these consequences threaten livelihoods, food security and economic stability.

7200AA20CA00022 as part of Feed the Future Innovation Lab for Animal Health. the funders had no role in the study design, data collection and analysis, decision to publish, or preparation of the manuscript.

**Competing interests:** The authors have declared that no competing interests exist.

Management of ECF includes methods of prevention and treatment, such as using chemical acaricides to control ticks and treating infected cattle using therapeutic drugs (*Buparvaquone and Parvaquone*) [3, 4]. However, these methods are neither very effective nor sustainable. The consistent use of acaricides is generally too expensive for smallholder cattle farmers [3], and ticks may develop resistance against them. Treating sick cattle using therapeutic drugs is often ineffective, as the method has to be administered in the early stages of infection [5]. Another prevention method is immunisation through the infection treatment method (ITM). The infection treatment method (ITM) provides a sustainable method of protecting cattle against ECF. The vaccination involves injecting the cattle with live *T.parva* parasites while administering an antibiotic known as oxytetracycline to prevent the infection, leading to lifelong immunity to similar, or related parasites [4]. However, despite the existence of ITM for over 40 years, adoption rates in Eastern and Southern Africa remain low [6].

Several factors have been attributed to this low adoption rate, including technical challenges, the high vaccine cost and limited access to veterinary services and infrastructure [5–11]. The challenges related to the costs of the vaccine and technical challenges are compounded by gaps in distribution and awareness, further hindering the widespread implementation of vaccination programs. The distribution and awareness of most livestock vaccines in Kenya involve collaborative efforts by various stakeholders ranging from vaccine distributors (local governments to veterinary officers operating private agro-veterinary businesses), vaccine deliverers, animal health providers at the county and sub-county level [10] in some cases NGOs and local leaders [12–16]. However, the distribution of ITM vaccines presents unique challenges; ITM requires specialized storage and handling, which many vaccine distributors and deliverers struggle to manage due to resource limitations. The vaccine is, therefore, not readily available for direct purchase by livestock keepers, further hindering access [7].

While studies have extensively focused on factors like accessibility, affordability, and acceptability of the vaccines [4, 5, 7, 9], these do not fully explore the broader social, cultural and environmental contexts that also play a role in adopting ITM. This study sought to address this gap by adopting a more holistic approach to understanding the factors influencing ITM adoption. To achieve this, the study employed the Socio-Ecological Model (SEM) to understand the complexities of ITM adoption. The core concept of SEM is that behaviour is influenced by multiple levels, including interpersonal (knowledge, beliefs and perceptions), interpersonal (socio-cultural), community, organizational and policy factors [17]. The SEM, therefore, provides a comprehensive framework for understanding the various and interacting determinants of behaviours. In addition, the SEM is crucial in developing comprehensive interventions that systematically target mechanisms of change at each level of influence [17].

By applying the SEM, this study captured the complexity of ITM adoption, moving beyond accessibility, affordability and availability to understand the broader context in which decisions are made. This approach provides a nuanced understanding of ITM adoption decisions, ultimately contributing to more effective and tailored strategies for ITM control in Narok and similar settings.

## Materials and methods

### Study site and population

The study was carried out in Narok South, a sub-county in Narok County. The site was selected based on its representation of distinct agroecological zones (AEZs), which offered varying conditions for livestock management and ECF prevalence. The area is divided into several AEZs: II, III, IV, V, VI and VII. Zones II and III (sub-humid to semi-humid), are characterised by cross-breed cattle, often grazed in enclosed environments. These zones are

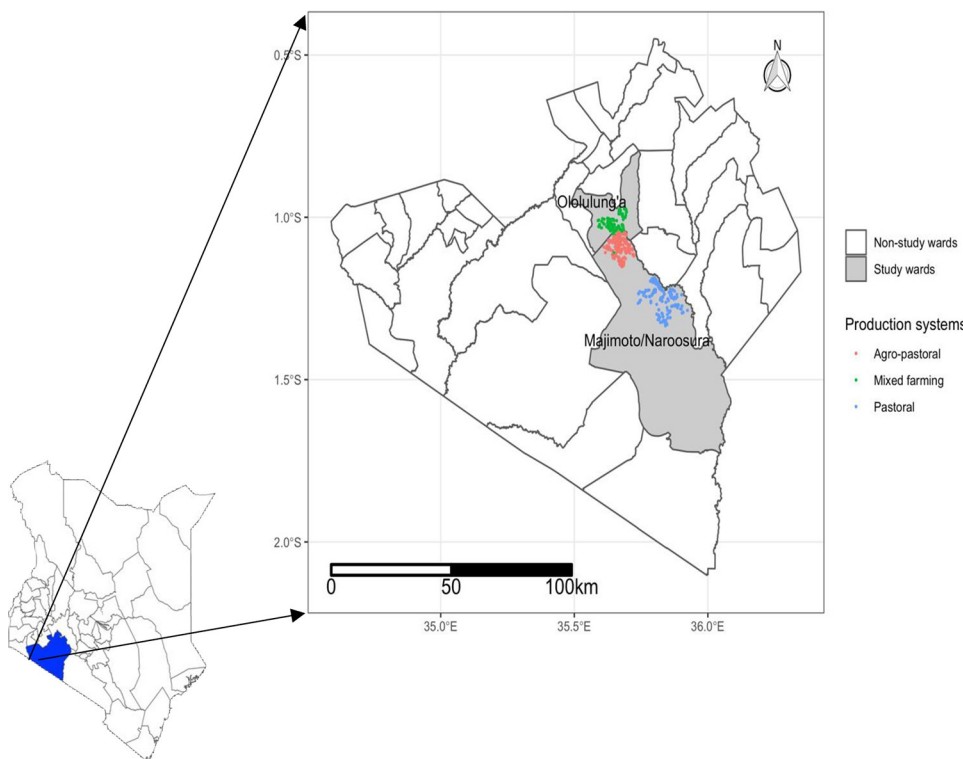

**Fig 1. Map of Kenya showing Narok County and specific study wards, the different production systems, and locations of the households where this study was carried out.** Source: The map was generated through GPS coordinates captured during the data collection process.

moderately suitable for ECF vectors, and the prevalence of FMD is low to moderate. Indigenous cattle breeds characterise Zone IV (semi-humid to semi-arid), and an open grazing system is practised. Agropastoralism is the main production system in this zone. The zone is suitable for ECF vector, and the prevalence of ECF is high. Indigenous cattle breeds characterise Zones V, VI and VII (semi-arid to very arid) and an open grazing system is practised. The zone is marginal to unsuitable for ECF vector, and ECF prevalence is low or non-existent. These zones are entirely livestock-dependent [5]. These zones significantly influenced livestock keepers' views and interactions with their environment and perception of ECF transmission.

Specifically, Naroosura Maji-Moto and Olololung'a wards in the Narok South sub-county were selected (**Fig 1**). These wards were selected based on their representation of different agro-ecological zones and production systems: pastoralism, agro-pastoralism and mixed farming.

Livestock keepers from twelve villages in Naroosura-Maji Moto and Olololung'a wards were selected for the study Table 1.

**Table 1. Production systems and corresponding villages in the Narok South sub-county.**

| Production system | Villages |
| --- | --- |
| **Agro pastoral** | Oloongila, Oloenae, Olkiriaine, Illadoru |
| **Pastoral** | Oldonyo-Orasha, Oloornga'nayio, Erupata, Inkasuriak, Nkintintini |
| **Mixed farmers** | Olepolos, Masaantare, Olgilai |

In the pastoral system, over 70% of livelihoods are derived from livestock kept on natural pastures, heavily dependent on livestock and their products. This system is characterized by seasonal migration, with livestock moving in search of water and pasture. By contrast, the agropastoral system practices crop cultivation and livestock keeping, with more than 50% of their income derived from crop cultivation [18]. In this system, crop residues are fed to livestock during the dry seasons, and when necessary, livestock are moved in search of water and pasture. In the mixed system, both crop cultivation and livestock keeping are practiced, with more than 70% of the income from crop cultivation [18]. The system is characterised by no migration of livestock. In these villages, elders play a vital role as channels for information regarding livestock disease outbreaks. The elders work with the community members to facilitate vaccination exercises during outbreaks, including resource mobilisation and coordinating vaccination schedules. The coordinated vaccination exercise occurs in a central location agreed upon by village elders.

The study population consisted of adults in the different villages who were males aged 18 years and above. The Maasai community's sociocultural setting, where men are traditionally given gendered duties related to livestock health and management, informed the selection of the male participants.

## Sampling and ethical clearance

Purposive sampling was used to recruit the study participants for the focus group discussions (FGDs), in-depth interviews (IDIs) as well as key informant interviews (KIIs). This sampling approach was selected deliberately to include individuals who could provide the most insightful and relevant information concerning our study objectives. Village elders from each of the selected villages helped with the mobilization of participants drawn from different villages.

All participants provided written informed consent. The tools and consent procedures were approved by the ethics review committee of the Kenya Medical Research Institute (KEMRI) and the National Council of Science, Technology, and Innovation (NACOSTI) clearance referenced KEMRI/RD/22 and NACOSTI/P/23/23342 respectively. Further, village-level authorities in the area also approved the study.

## Data collection

This was a focused ethnography to examine the determinants of ITM vaccine adoption among livestock keepers. Data was collected between April 2023 to December 2023. Various data collection methods were employed to examine perspectives from different stakeholders Table 2.

The FGDs provided insight into community and interpersonal levels of the SEM, capturing the social and cultural factors, including community attitudes and local norms in vaccine adoption. The FGDs lasted approximately 90 minutes, involving diverse groups of livestock keepers to ensure a broad representation of perspectives. The IDI participants, who were different from those in the FGDs, participated in interviews lasting between 45–60 minutes, exploring individual livestock keepers' beliefs, knowledge, perceptions and practices. Key

**Table 2. Data collection instruments.**

| Instrument | Location | Type of participant | Total |
|---|---|---|---|
| FGDs | 12 Villages | Livestock keepers | 18 |
| KIIs | Narok, Ewaso Ng'iro (towns) Oldonyo-Orasha, Olgilai, Illadoru, Masaantare, Olkiriane, Olepolos | Veterinarians and providers of agro vet services Local village elders and chiefs | 25 |
| IDI | 12 villages | Livestock keepers | 30 |

Informant Interviews (KIIs) lasted 30–45 minutes and were instrumental in understanding the SEM's sociocultural levels, offering experts' perspectives.

The data was collected by a multidisciplinary team of researchers and local animal health providers, including members familiar with the local context, language and cultural norms. This helped build rapport with participants, improve access to specific groups and facilitate open communication. To mitigate potential bias, a research team of two people per team, including both local animal health providers and external researchers, was formed. Research ethics training was also provided to all team members. Further, multiple data collection methods were used to cross-check and validate findings. This ensured the data that reflected the experiences of participants.

Interviews were audio recorded (with consent from participants) with the back-up of handwritten notes. Where participants did not agree to be recorded detailed handwritten notes were taken by the research assistants and typed for translation and analysis.

## Data management and analysis

The audio files were transcribed and translated from the Maasai or Swahili language into English by research assistants. Inconsistencies in translation were reduced through continuous discussions with translators and alignment of vocabulary and frequently used terms. The transcripts were then cross-referenced with corresponding notes to ensure accuracy and consistency. The transcribed data was then entered into NVivo $^{TM}$ (version 12) (NVivo QSR International) where a coding framework was created based on the SEM. As analysis progressed, sub-codes were identified from the initial codes. The emergence of the sub-codes allowed for a detailed analysis, aiding the identification of patterns and relationships that were integral to understanding the initial codes. Translated verbatim quotes have been used to illustrate the key points derived from the study.

## Results

### Demographic characteristics

A total of 130 participants (18–80 years of age) from twelve villages participated in the study. Over a quarter (28%) of the study participants were within the 26–33 years age range. Nearly half of the participants reported having no formal education Table 3

This study used a modified version of the socio-ecological model to include individual, socio-cultural, factors and ecological level factors as shown in Table 4.

Table 3. Socio-demographic characteristics of the participants.

| Category | Sub-category | Percentage (%) |
|---|---|---|
| Age | 18–25 years<br>26–33 years<br>34–41 years<br>42–49 years<br>>50 years | 23%<br>28%<br>18%<br>11%<br>20% |
| Education | None<br>Primary<br>Secondary<br>Tertiary | 46%<br>26%<br>21%<br>7% |
| Marital Status | Married<br>Single | 94%<br>6% |
| Religion | Traditional<br>Christian | 12%<br>88% |

**Table 4. Summary of factors influencing adoption of the Infection Treatment Method Narok South sub-county using the socio-ecological model.**

| SEM level | Factors | Summary |
|---|---|---|
| **Individual** | Knowledge and Awareness | Lack of awareness of the existence of ITM. Livestock keepers were aware of the existence of other livestock vaccines such as Foot and Mouth (FMD), Contagious bovine pleuropneumonia (CBPP), Contagious caprine pleuropneumonia (CCPP), and Peste des petits ruminants (PPR). |
| | Cost of ITM | Livestock keepers with fewer herd of cattle were willing to pay for the vaccine for all their herd unlike those with larger herds of cattle. |
| **Socio-cultural** | Community perception of ECF | Livestock keepers perceived certain breeds of cattle to be more susceptible to ECF. Indigenous cattle were perceived as less susceptible to ECF. Livestock keepers perceived that their areas were prevalent for ECF. Participants attributed this reduced prevalence to tick control measures, fencing of land and type of breed. Livestock keepers across the different production systems perceived other livestock diseases as more severe compared to ECF. Each production system had specific livestock diseases that were perceived to be more severe than ECF. |
| **Ecological** | Livestock-wildlife interaction | Livestock keepers perceived certain areas as being prone to ECF. This was attributed to the sharing of grazing land by both wildlife and livestock especially during migration. |
| | Climatic factors | Extreme weather conditions such as extreme heat or cold was also perceived to make certain areas more prone to ECF. Areas such as Mau and Mara were perceived as areas where ECF was prevalent. |

## Individual level factors

**Knowledge and awareness of the vaccine.** Across the different production systems, the majority (63%) of the livestock keepers were not aware of the existence of the ITM vaccine. However, a few (7%) across the pastoral and agropastoral systems had heard of the vaccine indirectly, mainly through discussion with livestock keepers from other regions where ECF was perceived to be severe.

*I have not heard of the vaccine for ECF I just know the drug used when the animal is sick* **(IDI-Pastoralists)**

*I have heard that the vaccine is being used in Mara and is very expensive. I have heard that it costs 1500ksh (11 $) per cattle*(**IDI-Agro-pastoralists).**

Village elders who were often responsible for mobilizing community members and disseminating information through local channels were also unaware of the vaccine as elaborated by this quote.

*We are not aware of a vaccine against ECF. Most times we organize vaccination exercises for foot and mouth disease (FMD, Contagious bovine pleuropneumonia (CBPP) and Contagious caprine pleuropneumonia (CCPP) (KII Village Elder)*

Animal health providers in these areas attributed the lack of awareness of the ITM vaccine by livestock keepers to several key factors among them the breed of cattle which was an important factor, with livestock keepers owning exotic breeds(imported cattle breeds) from other regions more likely to adopt the ITM. Furthermore, education, exposure and livestock keeper's experience with high cattle mortality rates from ECF were also attributed to the different levels of awareness. The following quotes by a local veterinarian demonstrate this:

*Livestock keepers in other places like Kajiado and Nakuru who have upgraded their cattle breeds and know their cattle are of high value tend to have information regarding ECF prevention measures, including vaccination. But in this area, very few know of the vaccine (**KII Vet Olololung'a)***

*In this area, those who know about this vaccine are educated and many of them are political leaders, people with great exposure. They have gone to different counties and seen what ECF vaccination is doing (**KII Vet Ewaso Ng'iro)***

Livestock keepers across the different production systems were however more familiar with vaccines for Foot and Mouth (FMD), Contagious bovine pleuropneumonia (CBPP), Contagious caprine pleuropneumonia (CCPP), and Peste des petits ruminants (PPR). This heightened awareness of other vaccines compared to the ITM was attributed to personal experience with the vaccines in the respective areas.

*I have not heard of the ECF vaccine. But I know of the vaccine that protects the cows against the common cold(CBPP). In 2022, we vaccinated our cows against it because there was an outbreak of the disease (**IDI-Pastoralists)***

*I have only heard of the vaccine for Olodua (PPR) and Olkirobi (FMD) because my livestock has been vaccinated recently (**IDI-Agro-pastoralists).***

Announcements from both the local radio stations and village elders were identified as sources of information for livestock keepers about livestock vaccines such as FMD, CBPP, and PPR. The information concerning these vaccines was particularly emphasized when there was a potential risk of outbreaks, thereby increasing awareness and ensuring that livestock keepers were organized for vaccination exercise within their areas. Village elders reported receiving timely information about the announcements of the outbreaks as demonstrated by this quote "*Just recently, there was an FMD outbreak in another village that is a bit far from here. I was alerted about this, and I had to organize other village elders to ensure that this information reached the community to ensure that the disease does not finish our livestock*" (**KII village elder Agro-pastoralists)**

An agro-veterinary service provider highlighted that similar efforts were undertaken to create awareness among livestock keepers for ECF as demonstrated in this quote "*There is a program in Sidai FM that I take part in. We usually advertise and sensitize farmers about FMD vaccination, we also advertise for the ECF vaccine (**KII Agro-vet-Olololung'a)***

However, livestock keepers and village elders reported not hearing of these announcements highlighting a communication gap.

**Price of the vaccine.** Given the lack of awareness about the ITM, a hypothetical scenario was presented to the livestock keepers about the possible adoption of the vaccine if it were commercially available at a price range of kshs 1000–2000 (8–15$) per cattle. This elicited diverse responses from the livestock keepers, with some expressing willingness to adopt the ITM, while others were not inclined to do so.

Across the different production systems, the majority (42%) of the younger livestock keepers (18–33 years) with relatively fewer herds (3–25 cattle) demonstrated a willingness to adopt the vaccine, emphasizing their focus on potential benefits over financial constraints. Similarly, a few (8%) older livestock keepers above 50 years who also had fewer herds (20–40 cattle) due to losses from drought expressed willingness to adopt the ITM vaccine. These livestock keepers contemplated the possibility of selling some of their livestock, such as cattle, sheep, and goats,

to finance the vaccination of the remaining cattle, which they perceived as a more effective approach than consistently treating infected cattle.

> *I will buy the vaccine even if it means selling the goats and using the money to buy the vaccine. The goats are 10,000kshs (75$), I will sell five and the money will even be more than enough to vaccinate all my cows* **(IDI Mixed Farmers)**.

However, most (28%) of the older livestock keepers aged 34–49 and above 50 expressed their willingness to adopt ITM, which was contingent upon the likelihood of recurring ECF outbreaks in their regions, as demonstrated in this quote "*If my cows are not infected with ECF then I will not vaccinate my cows but when one of my cows is infected, then I will have to vaccinate my cows. I will sacrifice one cow and sell it to vaccinate all my cows*" **(IDI-Agro-pastoralists)**.

Accessibility of information on ITM was also an important factor influencing the potential adoption of the vaccine. Livestock keepers with access to detailed information on the benefits and effectiveness were more likely to adopt the vaccine "*If I am told that the vaccine is effective and it prevents the cattle from being infected with Oltikana (ECF) and at what time I am supposed to vaccinate the cattle, I will buy the vaccine irrespective of the price*" **(IDI Agro-Pastoralists)**

Livestock keepers with larger herds of cattle particularly in the pastoral production system were reluctant to adopt the vaccine with the price of the vaccine posing a significant barrier as demonstrated by the quote "*If a person has 800 herds of cattle and the vaccination goes for 1000kshs(7$), for one cattle, the person will have to use 800,000kshs (6000$) to vaccinate all the cows and that is not possible. People will not afford that due to the high expense*"**(IDI -Pastoralists)**.

Within the mixed farming and agro-pastoral production system, there emerged a distinct preference for vaccine pricing to fall between 100 and 200 Kshs) ($0.76 to $1.51). In the pastoral production system, there was a preference for a much lower vaccine priced between 30-100kshs (($0.23 to $0.76).

> *Livestock keepers are not the same some have many cattle while others have just a few so the standard price should be 100ksh (0.76$)* **(FGD Agro-pastoralists)**

> *The vaccines should be sold at the same price as the other vaccines. Like we usually pay we usually pay 30kshs*($0.23) per cattle *for the vaccine that is administered at the tip of the cattle's tail (CBPP)* **(FGD Pastoralists)**

> *The vaccine should cost between 100–130 kshs ($0.76–0.91) to ensure that all livestock keepers can use it* **(FGD Mixed farmers)**

In addition to the price of the vaccine, the perception of ECF prevalence in their areas was a significant factor in the reluctance to adopt the ITM for livestock keepers. Due to the perceived lower risk, livestock keepers reported giving priority to other livestock vaccines considered prevalent in their areas as in the quote "*It is not that we will not vaccinate our cattle because of the price. Even if the vaccine costs 3000kshs (22$), we will still purchase and vaccinate our cows. But as I have said the reason I would not purchase the vaccine is because there are fewer cases of Oltikana (ECF). But if this was a vaccine against other diseases such as Olkipei(CBPP) and Olkirobi(FMD) where there are many cases in this area then I would have vaccinated my cows no matter the cost*"**(IDI-Pastoralists)**

## Socio-cultural level factors

**Social networks and informal communication channels.** Livestock keepers reported that knowledge about diseases and preventative measures often spread through social networks and informal communication channels. This information flowed through frequent interactions during informal gatherings such as market days, livestock trading centres and social visits.

*During the market days, we meet with livestock keepers from different places and we share information on the various livestock issues affecting each of our areas. When each of us goes back to our areas, we also share information that we have gotten from others (FGD Agro-pastoralist)*

*I first heard of the ECF vaccine when my friend from Mara came to visit. He told me that they use it there, but around here we sometimes don't see the need as our cattle are not so much affected by the disease (FGD Pastoralists)*

**Perception of ECF susceptibility among livestock keepers.** In all interviews, livestock keepers reported familiarity with ECF with a minority (24%) in the pastoral and agro-pastoral systems reporting encountering ECF in their herds, though highlighted a decline in its occurrence in recent years. While a few cases of ECF have been reported as recently as 2023, ECF appeared less prevalent than in previous years. Livestock keepers indicated that other diseases had a more significant impact on their herds.

*This year (2023, I have heard of very few cases of ECF being reported. However, the case is different with foot and mouth (FMD) as it is present throughout leading to the use of a lot of resources to control it (IDI-Pastoralists)*

*In 2021 when the drought began, that was when we had cases of ECF in our cattle, right now we are dealing with other diseases which are more common here(IDI Agro pastoralists)*

Livestock keepers attributed this decline to a variety of strategies implemented within their production systems. The consistent use of acaricides for tick control was emphasized in all interviews. Participants believed that regular application of acaricides to their cattle eradicated all ticks, including those that were responsible for ECF.

*They have been changes over the years because as time goes we have the strong dips(acaricides) that we use to spray the animals and this has reduced the tick infestation in the cattle reducing the chances of ECF in the cattle (FGD Agro-pastoralists)*

*The cases of Oltikana(ECF) were reduced because many people spray their cattle and manage the ticks. We were informed that the ticks cause ECF so, this made the people increase the frequency of spraying the cows, thus the cases reduced (FGD Pastoralists)*

A local veterinary doctor highlighted that ECF was easier to manage and control as it only needed proper tick control measures, as illustrated by this quote *"ECF is a manageable disease, and you can control it. We advise a schedule scheme to farmers. It is not always about vaccinations. You don't need drugs or antibiotics; you just need proper management of ticks. You can use Ivermectin, acaricides, and proper deworming so that you can control ECF (KII Vet Ololulung'a)*

Livestock keepers also noted that implementing fencing to prevent cattle interaction with wildlife or other cattle also contributed to the reduction of ECF.

*One of the causes of ECF is the interaction between wild animals and cattle. Presently, the establishment of wildlife conservations and the fencing of farms by individuals has led to a decrease in ECF cases in this area* **(FGD Pastoralists)**

*The cases have reduced because livestock keepers have fenced their farms, preventing interaction between the cattle***(FGD Agro-Pastoralists)**

The construction of water pans on livestock keepers' land was also identified as a contributing factor to the reduction in ECF cases. Livestock keepers noted that the water pans helped in managing the movement of livestock by providing a reliable water source *"Since everyone has the silanga (water pan) in their farms, each of us is usually cautious on the cattle that drink water in them and we make sure that other cattle do not drink water from our farms, thus managing the transmission of the disease"* **(FGD Pastoralists)**

In addition, participants in the agro-pastoral and pastoral production systems perceived that cattle breeds they reared were less susceptible to ECF. These perceptions were based on observations of how different cattle breeds responded to treatment when infected by ECF.

*The grade(imported) breed of cattle when they are infected with ECF will hardly get cured of the disease and in most cases die of it. However, the Indigenous breed(local breeds) which is sometimes resistant to the disease is cured of the disease when infected* **(FGD Agro-Pastoralists)**

**Perceived severity of ECF among livestock keepers.** Livestock keepers across the different production systems noted that while cases of ECF had affected cattle the last year, the symptoms of ECF in their region were not severe, and their cattle often recovered quickly. This was, however, different for other places where ECF was considered both prevalent and more severe as per this excerpt *"There is a type of ECF that the cows are infected with and it makes the cows produce bloody cow dung. This usually shows that the cows have been affected by the disease and they may die after two days of producing the bloody cows' dung. We usually call it, Sertet. This results in faster death of the cows. The type of Oltikana (ECF)in our area causes the cattle to have diarrhoea and this is usually manageable.* **(IDI-Pastoralists)**

This belief was rooted in the experiences of livestock keepers when their cattle were infected by ECF from both regions. According to their accounts, the recovery of cattle following treatment for ECF contributed to the perception that the disease was less severe in their localities.

Further, livestock keepers assessed the severity of ECF compared to other livestock diseases within their different production systems. The majority(56%) of participants across FGDs and IDIs expressed the perception that ECF was not deemed severe in comparison to other livestock diseases. However, each production system had specific livestock diseases that were perceived to be more severe than ECF. According to the livestock keepers, severe diseases were classified as those that result in higher mortality rates, affect a larger number of livestock, occur frequently, are unpredictable, and diseases for which treatment is not available as shown in Table 5.

However, about 10% of IDI respondents from both agro-pastoralist and pastoralist production systems perceived ECF to be more severe than other livestock diseases. This was attributed to how fast the cattle succumbed to ECF in contrast to other diseases and the cost of treatment as demonstrated by livestock keepers:

**Table 5. Summary of perceived severity of East Coast Fever compared to other livestock diseases in Narok South sub-county.**

| Dimension | Comparison | Quotes |
|---|---|---|
| Availability of treatment | Diseases with accessible treatments were perceived as less severe while those with unavailable treatments were perceived as more severe. | *If you wish to help farmers, you can help us deal with Ormilo (heartworm) which cannot be treated. However, with ECF I can easily get drugs to treat my cattle* **(FGD-Mixed farmers)** <br> *We don't have drugs to treat Shamsham (Bluetongue virus) and engati (Malignant Catarrhal Fever) unlike Oltikana (ECF) which treatment is readily available* **(FGD-Pastoralists)** <br> *We have drugs for Oltikana (ECF) but for Shamsham (Bluetongue), we do not have the medicine to cure the sheep and goats* **(FGD-Agro-pastoralists)** |
| Herd size affected | Livestock diseases with a significant impact on a larger number of livestock, were perceived as more severe | *Olkirobi (FMD) is the most contagious. It will infect many cattle and not one. It will spread to many cattle in a few days* **(FGD-Mixed farmers)** <br> *Oltikana(ECF) is better compared to Olodua(PPR) both with the livestock and to livelihoods because oltikana (ECF) is easy to treat and it doesn't affect all the cattle at once so treating the infected animal is not very expensive. But Olodua (PPR) is not treatable and it affects the cattle very much and sometimes all the cattle at once and it ends up killing other cattle* **(IDI-Agro-Pastoralists)** <br> *Olkirobi (FMD) and olodua (PPR) really affects us. They spread easily among many cattle at the same time* **(FGD-Pastoralists)** |
| Mortality | Diseases with high rate of mortality were seen as a threat to their livestock and livelihoods. | *Olodua (PPR) is a deadly disease and spreads easily. When the cows are infected with Oltikana(ECF), they die one cattle at a time but with PPR they die all at once* **(FGD-Pastoralists)** <br> *Olkipei(CBPP) has a higher death rate than Oltikana (ECF) since it affects both cattle and goats* **(FGD-Agro-Pastoralists)** <br> *When the cattle are infected with engati(MCF), the end result is death of many cattle at once* **(FGD-Pastoralists)** |
| Frequency of occurrence | Diseases that occurred frequently were viewed as more severe | *If Olkirobi (FMD), Olkipei (CBPP) and Olodua (PPR) are dealt with, I don't see other diseases being a problem. These are the diseases that keep reoccurring in these areas* **(FGD Mixed Farmers)** <br> *There are fewer cases of ECF than in the previous years. We can even stay for two years without our cattle getting infected with ECF* **(FGD Pastoralists)** <br> *The cows are not infected with Oltikana frequently, but they are mostly infected with Olkirobi* **(IDI Agro-Pastoralists)** |
| Unpredictable | Livestock diseases that farmers could anticipate when they would occur were perceived as less severe | *Shamsham (Bluetongue virus) is the most dangerous because we cannot tell when our sheep will be infected. However, with oltikana (ECF) we know that our cattle are more likely to get infected during migration* **(IDI Pastoralists)** |

*ECF is the most dangerous disease because if the animal is not treated early it will die within a very short time* **(IDI-Agro-pastoralists)**

*When the cattle are infected with FMD I may just administer Terramycin(Oxytetracycline) once the cows are cured but with ECF sometimes the medicine is not effective. You will have to administer the medicine even thrice so that the cattle can be treated* **(IDI Pastoralists)**

## Ecological factors

Two key ecological factors significantly influenced livestock keepers' perception of ECF, which could potentially affect the decision to adopt ITM. These factors were livestock grazing together with wildlife and weather conditions. However, these factors were not perceived in isolation but as part of a broader set of challenges livestock keepers faced in managing their herd health, especially during hunger and drought.

Across all IDIs and FGDs, participants in the agro-pastoral and pastoral production systems discussed the significant role that wildlife played in the transmission of ECF, particularly in regions where livestock share grazing areas with wild animals. Livestock keepers believed that wildlife were the carriers of many diseases including ECF which could potentially affect their cattle. These perceptions were further compounded by environmental stressors, such as

seasonal migrations and fluctuating weather conditions which livestock keepers believed exacerbated the risk of disease transmission.

*In the region such as Lemek and Ngosuani, there are wild animals that have ticks and the cattle graze in the same area where the wild animals are and this makes the animals have ECF every time* **(IDI-Agro-pastoralists)**

*I know there is a lot of ECF in Masai Mara because of the wild animals, but it's not prevalent in our area. We only have one or two cows that are affected over time* **(FGD Pastoralists)**

At the same time, weather conditions were perceived to contribute to the prevalence of ECF. Livestock keepers noted that specific areas with either extreme cold or dry conditions were prone to ECF, and these conditions also affected the cattle's resilience. This added a layer of complexity to ECF management, as livestock migrations to new areas for grazing could expose herds to unfamiliar weather conditions, which weakened their immune systems and increased vulnerability to ECF and other diseases. The quotes by livestock keepers exemplify this:

*When the cows migrate to new regions, like the colder areas of Mau, they encounter an environment that is significantly different from what they are accustomed to. The chilly climate of Mau, being unfamiliar to the cows, makes them susceptible to contracting Oltikana due to the stress of adaptation and the challenges posed by the colder conditions* **(FGD Agro-pastoralists)**

*When you go to areas such as Ntuka, it is usually drier than this area and the disease usually thrives in the dry conditions when many diseases affect the cattle during the dry season* **(IDI-Pastoralists)**

Livestock-wildlife interactions and weather played a crucial role in shaping livestock keepers' behaviour, particularly in times when there was a shortage of pasture and water for the livestock. These factors influenced the decisions on where to migrate cattle and the type of drugs to be used.

While livestock keepers recognized the higher prevalence of ECF in areas like the Mara and Mau region, few participants (14%) in the mixed farmers and pastoral system preferred migrating their cattle to Mau rather than Mara. This preference was due to the perception that ECF in Mau was easier to manage than Mara's. This is exemplified in this quote "*We take our cattle to Mau because we realized that the Oltikana (ECF) strain in that area is not as deadly as the one in Mara, it is the same strain as the one in this area*" **(FGD Agro Pastoral)**

Livestock keepers also noted that Mau had a lower incidence of other livestock diseases, which were also less severe than those in Mara. "*We prefer taking the cattle to Mau than Mara because in Mara there are many cases of Oltikana (ECF) and there are also many diseases. There are also Tsetse flies in Mara that sting the cows and also infecting them. Thus, we prefer taking the cattle to Mau*" **(FGD-Pastoralists)**

However, most livestock keepers (75%) across different livestock production systems expressed a different perspective. Participants reported that taking their cattle to Mara was better despite the perceived ECF risk. Livestock keepers noted that while ECF poses a significant health risk to their livestock, the immediate threat of starvation due to inadequate pasture presents a more dire consequence. Consequently, they believe that ECF's impacts can be mitigated through the administration of more potent drugs, should an outbreak occur.

*If there is grass in Mara, then we will take the cows to Mara even if there are many cases of Oltikana (ECF). Hunger can kill many cows, but ECF can be cured by giving stronger drugs* **(FGD -Pastoralists)**

*I understand the risks of ECF in Mara, but when there is drought, I must decide fast. It is about saving the cattle from drought. I can deal with ECF later if needed, but now pasture is my priority* **(FGD Agro-pastoralists)**

When migrating to the Mara region, livestock keepers used "stronger drugs" in the event the drugs they were used to did not work. In most cases, these farmers considered these drugs very expensive.

*When our cattle migrate to Lemek or Aitong we know that we need stronger drugs for ECF, but the problem is that these drugs are very expensive when many cattle are infected* **(FGD-Agro-pastoralists)**

*When we moved to Ngosuani in 2022, one of my cows got ECF. I tried administering Terramycin (oxytetracycline), but it was still not well. I bought another drug, and it cost me 2500kshs (18 $) to treat one cow* **(FGD-Pastoralists)**

## Discussion

This study applied the socio-ecological model (SEM) to identify multi-level factors influencing the adoption of the Infection Treatment Method (ITM). At the individual level, the study revealed that knowledge and awareness of the ITM were limited among most livestock keepers. This lack of awareness is critical, underscoring a significant barrier to ITM adoption. Similarly, a study among dairy farmers in several counties in Kenya highlighted that awareness of the ECF vaccine was low at 41%, attributing this to the low adoption rates [6]. Key informants in the study reported that awareness of ITM varied by livestock keepers' experience with high mortality rates from ECF, and the breed of cattle kept, with livestock keepers from other regions having greater awareness and demand for the vaccine. Conversely, since ECF was not viewed as a significant threat among livestock keepers in the study area, there was no motivation to seek information about the vaccine. This finding suggests that disease risk was assessed through the lens of lived experience and shared communal knowledge. Key informants also highlighted that livestock keepers in the region who were more exposed and educated were more likely to be aware of the vaccine. This shows that awareness was influenced by both individual risk perception and broader social and economic factors.

The limited awareness of ITM contrasts with livestock keepers' familiarity with other livestock vaccines such as FMD, CBPP, and PPR. The frequent and targeted vaccination campaigns for these diseases, necessitated by recurrent outbreaks, made these vaccines an integral part of livestock keeper's routine practices. However, the ITM vaccine was not actively promoted by local veterinarians or agro-vet service providers, making livestock keepers less aware of its existence. A study on the ECF vaccine among livestock keepers in Kajiado Central, Kenya, found that most were unaware of the vaccine. This lack of awareness was due to the assumption by animal health providers that livestock keepers would demand the vaccine without the need for promotion [9]. The findings underscore the need for campaigns to raise awareness about the ITM vaccine. Utilizing established communication channels and strategies used for other livestock vaccines could be effective in raising the livestock keepers' awareness about ITM's benefits and significance. Engaging influential community members such as chiefs and village elders to lead and develop awareness could enhance trust and encourage

broader vaccine adoption within the community. This study also underscores the significance of proactive and consistent awareness campaigns in influencing vaccine adoption rates. The findings also highlight the need to improve vaccine adoption and veterinary services to move from an outbreak-driven approach to a more consistent, preventative approach. Consistent outreach efforts and community involvement can raise awareness and build trust in vaccination initiatives and ultimately improve vaccine adoption.

In the study, cost was a significant barrier to the potential adoption of the ITM, with livestock keepers with more herds viewing vaccination of all their cattle as an expensive financial burden. The vaccine was viewed as expensive for many pastoralists who relied heavily on livestock, creating a barrier to widespread adoption. In contrast, distinct preferences for ITM pricing emerged between different production systems. Livestock keepers in the agropastoral and mixed farming production system were willing to pay slightly higher prices for the vaccine compared to the pastoralists. This could be linked to the diversification of income in the mixed and agropastoral production systems. In contrast, livestock keepers in the pastoral production system, due to their dependence on livestock as the sole source of income and the large size of the herds, made them less likely to invest in vaccines at higher costs. In addition to cost, the perception of ECF prevalence was also a significant factor shaping ITM adoption among these livestock keepers who prioritised other livestock vaccines for diseases that were perceived to be more prevalent in the area. Therefore, the decision-making process regarding ITM adoption was, therefore, influenced by not only cost but also the sociocultural understanding of disease risk. These findings contrast with a study on the adoption of the ECF vaccine among smallholder dairy farmers in North Rift Kenya, which found that off-farm occupation and herd dynamics were critical determinants in the decision-making process of ECF vaccine adoption [7]. Interventions to promote the adoption of ITM should tailor messages that align with the beliefs of the livestock keepers. This includes understanding how communities prioritise livestock diseases and shifting perceptions by contextualizing the ITM vaccine within livestock keepers' lived experiences.

At a sociocultural level, the study found that social networks played a significant role in spreading information about livestock diseases and management among livestock keepers in different regions. Livestock markets were key spaces where livestock keepers from different regions discussed challenges faced, such as livestock diseases. Similarly, another study among pastoral communities in Kenya identified social networks as crucial for facilitating the exchange of information about disease control measures. These networks allowed for the exchange of information across diverse geographical regions, allowing livestock keepers to share experiences [19]. Findings from the study revealed that few livestock keepers had heard about the ECF vaccine through informal channels. However, this did not translate to widespread vaccine knowledge among livestock keepers across the production systems. This highlights that information among livestock keepers was based on what was perceived as most relevant within a specific context. Since ECF was not perceived to be prevalent in the study area, livestock keepers did not prioritize discussions about the vaccine, resulting in limited awareness about ITM. While social networks are crucial in the sharing and exchanging of knowledge among livestock keepers, the relevance of the information is shaped by contextual factors. Understanding this is crucial for developing effective campaigns that align with the community.

Livestock keepers across the different production systems perceived that they could effectively manage ECF risk through their established practices, such as fencing of land and tick control methods, reducing the need for vaccination. Additionally, when cattle got infected, livestock keepers relied on readily available drugs, which they perceived as effective treatments. These findings align with a study on the Rift Valley Fever (RVF) vaccine uptake among

livestock keepers in Kenya and Uganda, which revealed a preference for curative services over vaccinations. This preference was due to the observation that the risk of an RVF outbreak was not always imminent, even when the livestock were vaccinated against it [20]. In addition, the perception that the breed of cattle kept in the study area was less susceptible to ECF further reduced the need for the adoption of ITM. Interventions should integrate the ITM vaccine into existing disease management practices. This could involve highlighting the vaccine as a complementary measure that enhances the effectiveness of other existing methods. The intervention should align with existing practices by demonstrating how ITM can work alongside these established practices.

The results of this study reveal that within the different production systems, most livestock keepers considered diseases other than ECF to be more severe. Diseases perceived as severe were those characterised by high mortality rates, frequency of outbreaks, diseases for which no available treatment options and predictability of diseases. Few livestock keepers perceived ECF as severe due to the treatment costs involved and how fast cattle succumbed to ECF in contrast to other diseases. These findings highlight the severity of diseases is culturally constructed based on the visible impact on livelihoods and livestock health. Diseases that lead to high mortality occurred frequently and affected a lot of livestock at the same time, posed an observable threat to livestock keepers' economic stability and social well-being. Furthermore, unpredictable diseases in terms of their onset were perceived as severe as they challenged livestock keepers' ability to prepare and respond effectively to disease threats.

Similarly, a study conducted in Ethiopia and Uganda, revealed that farmers prioritized diseases based on incidence, mortality rates, market value, reduced production, and treatment costs [21, 22]. Another study among pastoralists in Turkana, Kenya found that pastoralists believed that diseases with the highest prevalence and mortality rates, such as PPR, were perceived to be more important. Diseases perceived to have lower prevalence and mortality rates and that were not consistently present throughout the year or across different environments were considered low-priority [23]. Further, among pastoralists in Kajiado County, Kenya, diseases prioritized were those with no treatment options available, such as coenuruses, those that frequently occurred, like FMD and CBPP, and those that affected carcasses, therefore making them unfit for consumption. On the other hand, diseases that were perceived as easily treatable, such as bovine ephemeral fever, trypanosomiasis, and bloody diarrhoea, were treated as low-priority diseases [24]. Findings from this study reveal that interventions aimed at promoting the adoption of ITM should understand and align with local priorities. Tailoring ITM campaigns to align with livestock keepers, such as highlighting risks that may not be obvious, can ensure that the campaigns are relevant and resonate with the community, ultimately contributing to improved adoption rates.

From the study, mixed farmers were not exposed to the risks associated with migration as they did not move with their livestock in search of water and pasture. However, agro-pastoral and pastoral production systems demonstrated a specific pattern in managing their livestock movements, particularly concerning the perceived risk of ECF. Livestock keepers perceived the Mau and Mara regions as being more prevalent for ECF. These areas were considered to have a severe type of ECF compared to their areas, where ECF was less prevalent and easily treatable. Livestock keepers perceived that migration exposed cattle to ECF due to severe weather conditions and livestock-wildlife interactions. Despite acknowledging the risk of ECF during migration, livestock keepers prioritized avoiding starvation, which was viewed as a more immediate and severe threat. Starvation was perceived to have the potential to result in the death of many cattle at once, whereas ECF, though serious could be managed with more potent drugs. Similarly, findings from Uganda show that pastoralists faced a difficult decision in balancing the immediate need for water and pasture against the potential health risks posed

by diseases such as ECF [25]. Study findings show that social networks also significantly shaped the perception of disease risk and response. Frequent reports of ECF outbreaks, in the neighboring Mara ecosystem compared to their areas, led to the perception that those areas were more prone to the disease. This perception, shaped by the sharing of information through social networks, influenced the response to the risk of ECF. A study among pastoralists in Northern Kenya also highlighted that decision-making processes in response to animal diseases were influenced by different knowledge networks among them locally embedded networks to effectively share experiences and manage diseases [26].

## Conclusion

This study highlights the multifaceted factors affecting the adoption of livestock vaccines, demonstrating a complex interplay among individual, socio-cultural and ecological, level determinants.

Individually, key factors included the limited awareness of the ITM vaccine and its cost which was also linked to the perceptions of ECF severity. These factors play a significant role in the decision-making processes for adopting ITM, emphasizing the need to raise disease and vaccine awareness to change people's perceptions of susceptibility. The variance in perceptions also indicates the need for communication strategies tailored to individual experiences and concerns. Socioculturally, perception of ECF severity plays a large role in vaccination decisions and this could account for lower adoption rates of ITM vaccine in communities interpreting ECF as less severe. This study highlights the significance of culturally responsive awareness programs aligned with local beliefs and practices. Ecologically, the study highlights how climate conditions and livestock-wildlife interactions impact perceptions towards vaccination.

## Limitations

In this qualitative study, challenges inherent to qualitative research, such as biases and variations in data interpretation, were effectively mitigated through several rigorous methodological strategies. Key among these strategies was data triangulation, which involved the comparison of various data collection methods including in-depth interviews, focus groups, and observations, to ensure a comprehensive understanding of the research context. Further enhancing the reliability of our findings, the process of careful translation and back translation, coupled with a deep understanding of the cultural contexts in which data collection was conducted, played a crucial role.

## Supporting information

**S1 Dataset. Transcripts.**
(ZIP)

## Acknowledgments

We express our sincere gratitude to study participants in the Narok South sub-county for their contribution to the data presented in this article. We are grateful to the village elders who provided support and guidance during the data collection process. We also appreciate the research assistants involved in the process of data collection.

## Author Contributions

**Conceptualization:** Ann W. Muthiru.

**Data curation:** Ann W. Muthiru.

**Formal analysis:** Ann W. Muthiru, Nyamai Mutono, Salome A. Bukachi.

**Funding acquisition:** S. M. Thumbi.

**Investigation:** Ann W. Muthiru.

**Methodology:** Ann W. Muthiru.

**Project administration:** Ann W. Muthiru, Josphat Muema.

**Resources:** Josphat Muema.

**Supervision:** Salome A. Bukachi.

**Validation:** Ann W. Muthiru.

**Writing – original draft:** Ann W. Muthiru.

**Writing – review & editing:** Josphat Muema, Nyamai Mutono, S. M. Thumbi, Salome A. Bukachi.

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
