## [Decision Letter · Decision Letter 0]

8 Aug 2024

PONE-D-24-12806Beyond the Jab: Navigating Cultural Currents and Ecological Layers in East Coast Fever Vaccine Adoption in Rural Kenya.PLOS ONE

Dear Dr. Muthiru,

Thank you for submitting your manuscript to PLOS ONE. After careful consideration, we feel that it has merit but does not fully meet PLOS ONE’s publication criteria as it currently stands. Therefore, we invite you to submit a revised version of the manuscript that addresses the points raised during the review process. Please submit your revised manuscript by Sep 20 2024 11:59PM. If you will need more time than this to complete your revisions, please reply to this message or contact the journal office at plosone@plos.org. Please include the following items when submitting your revised manuscript:A rebuttal letter that responds to each point raised by the academic editor and reviewer(s). You should upload this letter as a separate file labeled 'Response to Reviewers'.A marked-up copy of your manuscript that highlights changes made to the original version. You should upload this as a separate file labeled 'Revised Manuscript with Track Changes'.An unmarked version of your revised paper without tracked changes. You should upload this as a separate file labeled 'Manuscript'.

We look forward to receiving your revised manuscript.

Kind regards,

Timothy Omara, PhD

Academic Editor

PLOS ONE

“This work was funded in whole or part by the United States Agency for International Development (USAID) Bureau for Resilience and Food Security under Agreement # 7200AA20CA00022 as part of Feed the Future Innovation Lab for Animal Health.”

Reviewers' comments:

Reviewer's Responses to Questions

**Comments to the Author**

1. Is the manuscript technically sound, and do the data support the conclusions?

Reviewer #1: Yes

Reviewer #2: Partly

2. Has the statistical analysis been performed appropriately and rigorously? 

Reviewer #1: Yes

Reviewer #2: N/A

3. Have the authors made all data underlying the findings in their manuscript fully available?

Reviewer #1: Yes

Reviewer #2: No

4. Is the manuscript presented in an intelligible fashion and written in standard English?

Reviewer #1: Yes

Reviewer #2: Yes

5. Review Comments to the Author

Reviewer #1: This is a good in-depth study based on a remarkable amount of data on socio-ecological factors influencing the perception of ITM for ECF of farmers in Kenya. Having researchers with in-depth knowledge and understanding of the context is invaluable.

General comments:

The manuscript can be shortened by avoiding repetition, particularly in the introduction. Going from the general situation to the context in Kenya and then the particular study sites would improve readability.

What is the availability of vaccines in Kenya and can you mention how vaccines and awareness strategies usually reach the farmers? Livestock field officers, agrovet and self-administration, government interventions??

As Plos One is a journal with a readership from many different disciplines, it would strengthen the manuscript if certain terms were introduced (such as SEM, exotic cattle, grand breed etc.) better.

Please add a paragraph on the study team and some more info on data collection such as length of interviews. You mention that deep understanding of context reduced bias, so please explain who collected the data.

In the results section if would help if there were some quantitative indicators (i.e. majority of farmers, minority, some, a few, 23%) as to how many interviewees mentioned particular issues and what characteristics they had such as “village chief”, “small herd owner” etc. to allow the reader to judge the importance of their contribution.

Consider a sentence with a short summary statement after each section, such as “Mara is better for grazing during droughts, but provides more interaction with wildlife”.

Table captions should be clear enough to be understood standing alone. Therefore, please do not use abbreviations in the captions and add site of study.

Table 2: To improve readability, please put a line between all columns and maybe rename the column “participants” ”type of participant”. Aligning all text to the left would also make the table clearer.

Table 3: please do not use 130%. N=130 which should stand for 100%. Recalculate the percentages in the table accordingly.

Table 4: No abbreviations in the caption please. Line 314 it should read “heads of cattle”.

Table 5: why is it in colour? Continuity of format across all tables. Ormilo is heartworm, not heartwater, please correct. In the section for unpredictable, please change received to perceived.

The discussion has to be changed, so that referenced studies are not just included as numbers where they are part of a sentence. For each study used as part of a sentence, study site and relevant finding has to be spelled out.

Do your findings suggest that higher positioned farmers (chiefs, elders, larger herd owners) have more knowledge, better knowledge? This cannot be easily assumed from reading the results section, but can be clarified in the results section as mentioned above and picked up in the discussion.

Add a sentence on how influential community members could be used to lead and develop campaigns.

Specific comments:

Intro:

Please clarify:

Deaths are part of losses – reword

Typically HH with limited resources are affected – why?

Exotic cattle?

Economic losses – who and why?

Reduce repetition by starting with overall explanation of the complex interactions in East Africa and going to the specific situation in your study site. Make sure the described situation is more clearly applied to Kenya and your study population.

l. 289, replace “a few” with a number if possible.

L 357 last year is not “a few years ago”!

L 385 explain grade breed

L 399 to have diarrhea

L429 in 4 IDIs

L 600 explain ref 6 to complete the sentence like in l 609

L 618 punctuation

L 660-664 any comparison with other studies? Any reason why?

683 ECF tends to impact only smaller herds – why? Do you mean “the impact of ECF is felt stronger by owners of smaller herds”?

Reviewer #2: Beyond the Jab

General

Title of the paper; Wordy but unclear and not representative of the contents of the article; needs revision

As this is a qualitative study, it would be useful to describe the characteristics of the livestock keepers. For example, what kinds of people were mixed farmers or pastoralists and how are they differentiated in terms of age, education status? This is especially so because the paper clearly distinguishes between the perspectives of these different categories of farmers. What is mixed farming vs agro pastoralism?

In the introduction, the authors describe vaccination. It would be helpful to know if this disease is treatable and if the treatment is easily available? If treatment exists is it cheaper than the vaccines for example?

The study sites were selected based on contextual and ecological considerations. What were they? We do not see in this paper what this consideration for context and ecology provided in terms of enriching the data. For example, did those living close to wildlife areas discuss more the ecological aspects of SEM than those located in other areas?

In the results section, there is barely any contribution by the key informants to the study.

References missing, fullstops missing,

Not clear what the differentiation among the production systems is supposed to achieve? In the discussion section, it does not come up and in the results section barely.

82-83 Repetition as already stated in paragraph above

90-91 Already stated above

116 Narok South Subcounty?

117-118 Briefly explain the contextual and ecological considerations

124 Table 1; What is the difference between agro pastoral and mixed farmers?

134 Are the elders the same people described as community leaders?

138 Gendered duties not gender duties

163 KIIs You describe animal health care service providers (both county and private) then talk about animal health service providers in agro vets. How different is this latter category from “private practioners earlier mentioned? Were IDI participants different from FGD participants? Were the FGDs conducted according to the method of livestock farming ie pastoralists, agro pastoralists and mixed farmers? It appears so in the results but not mentioned in the methodology

180 Transition or translation?

184 Fullstop missing

247-248 Who were the livestock keepers who owned mixed breeds of cattle and thus more likely to adopt ITM? Were they people of a certain education or economic status?

249-250 See comment above 247-248 are describe if there is any correlation

259-260 The last part of this quote does not relate to the kind of exposure described in the two quotes. I see two different kinds of exposure: exposure to the vaccine and exposure to ECF and its impacts

354-360 This paragraph is unclear. Cases of ECF are reported to be few and to have occurred a few years ago but the quote is about ECF cases the year before the study? For clarity this could be compared with the occurrence of other diseases. For example, ECF might have occurred but maybe not as impactful as other livestock diseases?

385 what does “the grade breed of cows” mean? Exotic breeds?

454-455 The description is about the weather and not climate

470 the quotes above describe this. This part needs to be restructured for clarity and to avoid repetition

493-530 This section is unclear. I think it would be better to not narrow down to one issue at a time eg migrating to Mara vs Mau as this is a simplifying a complex issue. It seems that as exemplified by the majority in line 506-507 that the livestock keepers are describing the complexity of mitigating against the ECF disease and other diseases in a context of hunger and drought. The narrative here simplifies this issue unhelpfully.

534 Policy factors; how is the focus of vaccination campaigns and the information available a policy factor? These would be if in the introduction section it is clarified that the policy direction in the country is towards some livestock diseases and not others.

584-586 References?

608-609 Not included in the results but discussed?

609-611 Reference?

615 grade breeds of cattle? Meaning

621-623 Suggest rephrasing because pastoralists for clarity; do pastoralists know or expected to know the scientific evidence?

640-642 Reference?

689-691 are policy issues synonymous with public awareness campaigns?

708 Limitations after conclusion

719 Conclusion; Indeed, these issues are multifaced and there is complex interplay. However, the way they are discussed in this paper does not bring this out. It appears each factor is independent of the rest. This could have been mitigated through:

• More contextual insights of the study sites, livelihoods, study setting, why the different production methods were selected and what show value to the study,

• In the discussion section do not simply describe results and available literature but elucidate the interrelationships between and among the variables

6. PLOS authors have the option to publish the peer review history of their article (what does this mean?). If published, this will include your full peer review and any attached files.

Reviewer #1: No

Reviewer #2: No

---

## [Author Response · Author response to Decision Letter 0]

11 Nov 2024

Editor comments:

Comment: Please ensure that your manuscript meets PLOS ONE's style requirements, including those for file naming. 

Response: We have formatted the document as per PLOS ONE guidelines

Comment: Please state what role the funders took in the study.

Response: We have included a statement that the funder had no role to play in design, data collection and analysis, decision to publish, or preparation of the manuscript. 

Comment: Please include this amended Role of Funder statement in your cover letter

Response: We have included this in the cover letter

Comment:Please provide a complete Data Availability Statement in the submission form, ensuring you include all necessary access information or a reason for why you are unable to make your data freely accessible.

Response: We understand PLOS’ data policy and have ensured compliance by including selected transcripts as supporting documents. 

Comment: You may seek permission from the original copyright holder of Figure 1 to publish the content specifically under the CC BY 4.0 license

Response: We (the authors) created the map (Figure 1), and we hereby publish it under the Creative Commons Attribution License (CC BY 4.0). This license allows unrestricted use, distribution, and reproduction in any medium, provided the original author(s) and source are credited.

Reviewer 1:

Comment: The manuscript can be shortened by avoiding repetition, particularly in the introduction. Going from the general situation to the context in Kenya and then the particular study sites would improve readability.

Response: The introduction part has been revised to address these comments. It highlights ECF and impacts on livestock farmers in Africa; the presence of the ECF vaccine; low adoption rates in Southern Africa, Eastern Africa and Kenya; distribution and awareness of vaccines in Kenya; study gap and objective. Lines 58-137

Comment: What is the availability of vaccines in Kenya and can you mention how vaccines and awareness strategies usually reach the farmers? Livestock field officers, agrovet and self-administration, government interventions??

Response: This statement has been included to highlight the different stakeholders involved for livestock vaccines reach famers and various awareness strategies employed. It also highlights the unique challenges ITM presents in relation to distribution and awareness creation. Lines 107-114

Comment: As Plos One is a journal with a readership from many different disciplines, it would strengthen the manuscript if certain terms were introduced (such as SEM, exotic cattle, grade breed etc.) better. 

Response: We have addressed this by providing clear definitions and explanations for terms. Lines 120-126, 889

Comment:Please add a paragraph on the study team and some more info on data collection such as length of interviews. You mention that deep understanding of context reduced bias, so please explain who collected the data.

Response: This has been incorporated to provide details about the study team and how the team was selected to reduce bias. Additionally, information on the duration of the interviews has been included. Lines 349-378

Comment: In the results section if would help if there were some quantitative indicators (i.e. majority of farmers, minority, some, a few, 23%) as to how many interviewees mentioned particular issues and what characteristics they had such as “village chief”, “small herd owner” etc. to allow the reader to judge the importance of their contribution.

Response: Percentages have been used as quantitative indicators. In addition, characteristics of the interviewees have been highlighted. Lines 477-1150

Comment: Table captions should be clear enough to be understood standing alone. Therefore, please do not use abbreviations in the captions and add site of study.

Response: We have revised the table captions to remove abbreviations and added the site of study to ensure that the captions are clear and can be understood independently. Line 462

Comment: Table 2: To improve readability, please put a line between all columns and maybe rename the column “participants” ”type of participant”. Aligning all text to the left would also make the table clearer. Lines 346

Table 3: please do not use 130%. N=130 which should stand for 100%. Recalculate the percentages in the table accordingly.

Response: This has been addressed . Line 445

Comment: Table 4: No abbreviations in the caption please. Line 314 it should read “heads of cattle”.

Response: Abbreviations have been removed from the caption. Line 462

Comment: Table 5: why is it in colour? Continuity of format across all tables. Ormilo is heartworm, not heartwater, please correct. In the section for unpredictable, please change received to perceived.

Response: This table has been revised and format is the same as other tables. The other comments have also been addressed. Line 971

Comment: The discussion has to be changed, so that referenced studies are not just included as numbers where they are part of a sentence. For each study used as part of a sentence, study site and relevant finding has to be spelled out.

Response: The discussion section has been reworked to incorporate this comment. Lines 1156-1540

Comment: Do your findings suggest that higher positioned farmers (chiefs, elders, larger herd owners) have more knowledge, better knowledge? This cannot be easily assumed from reading the results section, but can be clarified in the results section as mentioned above and picked up in the discussion.

Response: This has been clarified in the result and discussion section. Lines 496-549, 1163-1172

Comment:Add a sentence on how influential community members could be used to lead and develop

Response: We have added a sentence to show influential community members could play a role in adoption of ITM . Lines 1264-1266

Comment: Deaths are part of losses – reword

Response: This statement has been improved. Line 61

Comment: Typically HH with limited resources are affected – why?

Response: This statement has been improved to highlight that livestock dependent communities are the most affected due to loss of livelihoods, food security and economic stability. Line 63-64

Comment: Reduce repetition by starting with overall explanation of the complex interactions in East Africa and going to the specific situation in your study site. Make sure the described situation is more clearly applied to Kenya and your study population. 

Response: The introduction section has been reworked to improve this. Lines 58-137

Comment: L 357 last year is not “a few years ago

Response: We have addressed by revising the text to accurately reflect the timeframe mentioned, clarifying that the most recent cases of ECF were within the past year. Line 769

Comment: 660-664 any comparison with other studies? Any reason why?

Response: We have addressed this comment by incorporating a comparison with another study which examines how pastoralists in northern Kenya use different knowledge networks to manage animal diseases under highly variable conditions. Lines 1315-1320

Comment:683 ECF tends to impact only smaller herds – why? Do you mean “the impact of ECF is felt stronger by owners of smaller herds”?

Response: We have addressed this comment by clarifying that ECF typically affects only a few animals within a herd, as opposed to other livestock diseases that can impact many animals or entire herds at once. Line 1166

Reviewer 2:

Comment: Title of the paper; Wordy but unclear and not representative of the contents of the article; needs revision

Response: The title of the paper has been revised and reflects the contents of the article. Line 1

Comment: As this is a qualitative study, it would be useful to describe the characteristics of the livestock keepers. For example, what kinds of people were mixed farmers or pastoralists and how are they differentiated in terms of age, education status? This is especially so because the paper clearly distinguishes between the perspectives of these different categories of farmers. What is mixed farming vs agro pastoralism

Response: We have added a description of the characteristics of pastoralists, agro-pastoralists, and mixed farmers, based on prior studies and our own observations. Lines 271-292

Comment: In the introduction, the authors describe vaccination. It would be helpful to know if this disease is treatable and if the treatment is easily available? If treatment exists, is it cheaper than the vaccines for example?

Response: We have revised the introduction to include information on the treatment options for ECF and the cost comparison with vaccination. Line 66-76

Comment:The study sites were selected based on contextual and ecological considerations. What were they? We do not see in this paper what this consideration for context and ecology provided in terms of enriching the data. For example, did those living close to wildlife areas discuss more the ecological aspects of SEM than those located in other areas?

Response: We have revised the manuscript to elaborate on the contextual and ecological considerations that guided the selection of the study sites and how these factors enriched the data. The study sites were selected to capture diverse agro-ecological zones and production systems—pastoralism, agro-pastoralism, and mixed farming—each offering different contextual and ecological dynamics. Lines 145-292

Comment: In the results section, there is barely any contribution by the key informants to the study.

Response: In response to your comment, we have now added more quotes and insights from the key informants to better reflect their contributions to the study. Lines 492, 542, 547,590,596,842

Comments: Not clear what the differentiation among the production systems is supposed to achieve? In the discussion section, it does not come up and in the results section barely.

Response: We have clarified this in the results and discussions. Lines 703, 713, 1004,1272,1452,1533

Comment: 82-83 Repetition as already stated in paragraph above

Response: The introduction section has been re written. Lines 57-137

Comment: 124 Table 1; What is the difference between agro pastoral and mixed farmers?

Response: We have highlighted difference has been highlighted in terms of livestock management practices and livelihoods derived from the different practices. Lines 271-292

Comment: 134 Are the elders the same people described as community leaders?

Response: Yes, the elders referred to are the same individuals described as community leaders. To maintain consistency throughout the study, we have adopted the uniform term 'village elders' to refer to them. Whole document

Comment: 138 Gendered duties not gender duties

Response: This has been rectified. Line 300

Comment: 163 KIIs You describe animal health care service providers (both county and private) then talk about animal health service providers in agro vets. How different is this latter category from “private practioners earlier mentioned? 

Response: We have used the general term 'animal health providers' throughout. However, in the Results section, we have specified the type of provider, distinguishing between agro vet shop owners and veterinary professionals who operate outside of agro vets, to provide greater clarity. Whole document

Comment: Were IDI participants different from FGD participants? Were the FGDs conducted according to the method of livestock farming ie pastoralists, agro pastoralists and mixed farmers? It appears so in the results but not mentioned in the methodology

Response: Participants for the IDIs and FGDs were different, but their selection was based on the specific livestock production systems. The methods were chosen to collect data tailored to the type of information, with FGDs or IDIs used depending on the appropriateness for the data required. This has been clarified . Lines 349-356

Comment: 180 Transition or translation?

Response: This has been rectified to translation. Line 388

Comment: 247-248 Who were the livestock keepers who owned mixed breeds of cattle and thus more likely to adopt ITM? Were they people of a certain education or economic status?249-250 See comment above 247-248 are describe if there is any correlation

Response: The livestock keepers who owned exotic breeds of cattle were from regions outside the study sites. These individuals were not part of the communities where the study was conducted, as the local cattle population primarily consisted of indigenous breeds and cross breeds. This statement has been added. Lines 496-549

Comment: 259-260 The last part of this quote does not relate to the kind of exposure described in the two quotes. I see two different kinds of exposure: exposure to the vaccine and exposure to ECF and its impacts.

Response: We have revised the manuscript and removed the last part of the quote to align with the type of exposure being discussed. Lines 457-459

Comment: 354-360 This paragraph is unclear. Cases of ECF are reported to be few and to have occurred a few years ago but the quote is about ECF cases the year before the study? For clarity this could be compared with the occurrence of other diseases. For example, ECF might have occurred but maybe not as impactful as other livestock diseases?

Response: We have addressed the comment by clarifying the timeline of ECF occurrences and providing a comparison with other livestock diseases. The revised paragraph now emphasizes that while ECF is still present, it is perceived as less of a concern compared to other more impactful diseases. Additionally, we have included relevant quotes to illustrate this distinction. Lines 767-779

Comment: 385 what does “the grade breed of cows” mean? Exotic breeds?

Response: Grade breed in the context of this study also means imported cattle. We have clarified that exotic cattle mean imported cattle breeds. Line 890

Comment: 470 the quotes above describe this. This part needs to be restructured for clarity and to avoid repetition.

Response: This has been revised. Quotes have remained but the explanation has been removed. Lines 1029-1034

Comment: 454-455 The description is about the weather and not climate

Response: This has been revised. lines 1000, 1018

Comment : 493-530 This section is unclear. I think it would be better to not narrow down to one issue at a time eg migrating to Mara vs Mau as this is a simplifying a complex issue. It seems that as exemplified by the majority in line 506-507 that the livestock keepers are describing the complexity of mitigating against the ECF disease and other diseases in a context of hunger and drought. The narrative here simplifies this issue unhelpfully.

Reponse: We have revised this section to reflect the complexity of the challenges livestock keepers face, particularly in managing ECF and other diseases within the broader context of hunger, drought, and environmental stress. We removed any oversimplification of specific issues and restructured the narrative to better capture the interconnected factors affecting disease transmission and cattle health, as suggested. Lines 998-1073

Comment: 534 Policy factors; how is the focus of vaccination campaigns and the information available a policy factor? These would be if in the introduction section it is clarified that the policy direction in the country is towards some livestock diseases and not others.

Response: We have moved the results and discussion around information to the section on knowledge and awareness of the vaccine. Lines 586-600

Comments: 689-691 are policy issues synonymous with public awareness campaigns?

Response: Policy issues and public awareness campaigns are distinct. We have removed the sub section of policy factors and moved the contents to individual factors

Comments: 719 Conclusion; Indeed, these issues are multifaced and there is complex interplay. However, the way they are discussed in this paper does not bring this out. It appears each factor is independent of the rest. This could have been mitigated through:

Response: The updated discussion now provides a deeper analys

---

## [Decision Letter · Decision Letter 1]

26 Nov 2024

PONE-D-24-12806R1

Beyond the jab: unravelling the complexities of vaccine adoption for East Coast Fever in rural Kenya

PLOS ONE

Dear Dr. Muthiru,

Thank you for submitting your manuscript to PLOS ONE. After careful consideration, we feel that it has merit but does not fully meet PLOS ONE’s publication criteria as it currently stands. Therefore, we invite you to submit a revised version of the manuscript that addresses the points raised during the review process.

We look forward to receiving your revised manuscript.

Kind regards,

Timothy Omara

Academic Editor

PLOS ONE

Journal Requirements:

Reviewers' comments:

Reviewer's Responses to Questions

**Comments to the Author**

1. If the authors have adequately addressed your comments raised in a previous round of review and you feel that this manuscript is now acceptable for publication, you may indicate that here to bypass the “Comments to the Author” section, enter your conflict of interest statement in the “Confidential to Editor” section, and submit your "Accept" recommendation.

Reviewer #1: All comments have been addressed

Reviewer #2: All comments have been addressed

2. Is the manuscript technically sound, and do the data support the conclusions?

Reviewer #1: Yes

Reviewer #2: Yes

3. Has the statistical analysis been performed appropriately and rigorously? 

Reviewer #1: Yes

Reviewer #2: Yes

4. Have the authors made all data underlying the findings in their manuscript fully available?

Reviewer #1: Yes

Reviewer #2: Yes

5. Is the manuscript presented in an intelligible fashion and written in standard English?

Reviewer #1: Yes

Reviewer #2: Yes

6. Review Comments to the Author

Reviewer #1: Very well revised manuscript and much improved. Comments of reviewers were addressed well. Some minor issues remain before it can be accepted:

1. There is a confusion between herds of cattle and heads of cattle.

A herd is a collection/group of animals. If someone has several herds, it means they have several distinct groups of cattle.

Heads of cattle are individual animals. Owning 5 heads of cattle means 5 animals.

Table 4 says “livestock keepers with fewer herd of cattle” – I think you mean heads of cattle.

L 362 in the quote you say - if a person has “800 herds of cattle” – I think you mean “800 heads of cattle”, i.e. 800 animals.

2. There is a confusing use of abbreviations for the disease and the vaccine against the disease.

For example, in L 107 FMD refers to the disease, while in table 4 FMD refers to the vaccine against the disease. Please go through the whole document and make sure this is clarified everywhere.

In Table 4 write: “existence of other livestock vaccines for FMD, CBPP …”

ECF vaccine or ITM vaccine? The Infection Treatment Method is a vaccine against ECF. Correct throughout so it is clear please.

Minor issues:

L 58 only define ITM once!

L 107 define FMD

L 128 – repetition in sentence – replace second livestock with animals.

L133 as above

L 201 remove “that”

L 204 add type of consent (written)

L 335 do you mean smaller herds?

L 376 correct repetition.

L 384 add “lower risk of EFC” “to other livestock vaccines for diseases considered prevalent”

L386 add “purchase it”

L 396 replace “flowed” with “spread”

L 427 replace “they” with “there”

L 447 what do you mean by “wildlife conservations” do you mean conservation areas?

L 455-458 revise sentence – makes no sense

L 564-581 What did the remaining 11% do? 14% prefer Mau, 75% prefer Mara and 11% prefer????

L 623 insert vaccines such as “for” FMD, CBPP….

Reviewer #2: This review is better and captures the comments earlier raised. It is important to demonstrate that the issues discussed are complex and intertwined so that the analysis is not too simplistic. Consider providing some relevant recommendations following this study.

7. PLOS authors have the option to publish the peer review history of their article (what does this mean?). If published, this will include your full peer review and any attached files.

Reviewer #1: No

Reviewer #2: No

---

## [Author Response · Author response to Decision Letter 1]

30 Nov 2024

Editor: Please note that funding information should not appear in the Acknowledgments section/Funding section or Any other areas of your Manuscript. We will only publish funding information present in the Funding Statement section of the online submission form. Please remove any funding-related text from the manuscript.

Response:

Funding related text has been removed from the manuscript

Reviewer 1

comments:

There is a confusion between herds of cattle and heads of cattle.

Table 4 says “livestock keepers with fewer herd of cattle” – I think you mean heads of cattle.

L 362 in the quote you say - if a person has “800 herds of cattle” – I think you mean “800 heads of cattle”, i.e.

800 animals.

Response: 

We have thoroughly reviewed the manuscript and made the following corrections

 In Table 4, we have replaced "livestock keepers with fewer herd of cattle" with "livestock keepers with fewer heads of cattle."

In Line 362, within the quote, we have corrected "800 herds of cattle" to "800 heads of cattle," reflecting that it refers to 800 individual animals.

Comments:

There is a confusing use of abbreviations for the disease and the vaccine against the disease.

For example, in L 107 FMD refers to the disease, while in table 4 FMD refers to the vaccine against the disease.

Please go through the whole document and make sure this is clarified everywhere.

In Table 4 write: “existence of other livestock vaccines for FMD, CBPP ...”

ECF vaccine or ITM vaccine? The Infection Treatment Method is a vaccine against ECF. Correct throughout so it

is clear please.

Response:

We have ensured that abbreviations for diseases and their vaccines are clearly distinguished throughout the manuscript.

When referring to the disease, we use the abbreviation alone (e.g., 'FMD' for Foot-and-Mouth Disease).

 When referring to the vaccine, we specify 'FMD vaccine' to indicate that it is the vaccine against the disease.

In Table 4, we have revised the text to read: 'Existence of other livestock vaccines for FMD, CBPP...' This explicitly indicates that we are discussing vaccines for these diseases.

We have corrected all instances for clarity

Comments:

L 107 define FMD

Response:

We have added the definition of FMD (Foot-and-Mouth Disease) at its first mention in the manuscript

Comments:

L 128 – repetition in sentence – replace second livestock with animals.

Response:

We have replaced the second occurrence of 'livestock' with 'animals' to avoid repetition.

Comments:

L 204 add type of consent (written)

Response:

We have specified in the manuscript that written consent was obtained from all participants.

Comments:

L 335 do you mean smaller herds?

Response:

We meant 'smaller herds'. We have updated the text to reflect this accurately.

Comments:

L 384 add “lower risk of EFC” “to other livestock vaccines for diseases considered prevalent”

Response: We have revised the sentence to include 'lower risk of ECF' and specified 'to other livestock vaccines for diseases considered prevalent' for clarity.

Comments:

L 564-581 What did the remaining 11% do? 14% prefer Mau, 75% prefer Mara and 11% prefer????

Response: The remaining 11% chose not to migrate their cattle to either Mau or Mara but opted to purchase hay to feed their livestock locally. We have updated the manuscript to include this information for clarity.

Comments:

L 623 insert vaccines such as “for” FMD, CBPP....

Response:

We have inserted the word 'for' to read: 'vaccines such as for FMD, CBPP...'

Comments:

L 455-458 revise sentence – makes no sense

Response:We have revised the sentence for clarity

Comments:

L 396 replace “flowed” with “spread”

Response: 

We have replaced 'flowed' with 'spread' for better readability.

---

## [Editor Report · Decision Letter 2]

3 Dec 2024

Beyond the jab: unravelling the complexities of vaccine adoption for East Coast Fever in rural Kenya

PONE-D-24-12806R2

Dear Dr. Muthiru,

We’re pleased to inform you that your manuscript has been judged scientifically suitable for publication and will be formally accepted for publication once it meets all outstanding technical requirements.

Kind regards,

Timothy Omara

Academic Editor

PLOS ONE
---

## [Editor Report · Acceptance letter]

16 Dec 2024

PONE-D-24-12806R2 

PLOS ONE

Dear Dr. Muthiru, 

I'm pleased to inform you that your manuscript has been deemed suitable for publication in PLOS ONE. Congratulations! Your manuscript is now being handed over to our production team.

Kind regards, 

on behalf of

Dr. Timothy Omara 

Academic Editor

PLOS ONE